# Structural and energetic profiling of SARS-CoV-2 receptor binding domain antibody recognition and the impact of circulating variants

Rui Yin[1,2], Johnathan D. Guest[1,2], Ghazaleh Taherzadeh[1,2¤], Ragul Gowthaman[1,2], Ipsa Mittra[1,2], Jane Quackenbush[1,2], Brian G. Pierce[1,2]*

**1** University of Maryland Institute for Bioscience and Biotechnology Research, Rockville, Maryland, United States of America, **2** Department of Cell Biology and Molecular Genetics, University of Maryland, College Park, Maryland, United States of America

¤ Current address: Department of Mathematics and Computer Science, Wilkes University, Wilkes-Barre, Pennsylvania, United States of America

* pierce@umd.edu

**Data Availability Statement:** Data are available from the CoV3D database, at these pages: https://cov3d.ibbr.umd.edu/antibody_classification https://

## Abstract

The SARS-CoV-2 pandemic highlights the need for a detailed molecular understanding of protective antibody responses. This is underscored by the emergence and spread of SARS-CoV-2 variants, including Alpha (B.1.1.7) and Delta (B.1.617.2), some of which appear to be less effectively targeted by current monoclonal antibodies and vaccines. Here we report a high resolution and comprehensive map of antibody recognition of the SARS-CoV-2 spike receptor binding domain (RBD), which is the target of most neutralizing antibodies, using computational structural analysis. With a dataset of nonredundant experimentally determined antibody-RBD structures, we classified antibodies by RBD residue binding determinants using unsupervised clustering. We also identified the energetic and conservation features of epitope residues and assessed the capacity of viral variant mutations to disrupt antibody recognition, revealing sets of antibodies predicted to effectively target recently described viral variants. This detailed structure-based reference of antibody RBD recognition signatures can inform therapeutic and vaccine design strategies.

## Author summary

The ongoing COVID-19 pandemic, and the emergence of SARS-CoV-2 variants that evade antibodies induced by vaccines and natural infection, highlights the need for assessment of key molecular and structural features of immune responses against the SARS-CoV-2 virus. Using a large nonredundant set of structures of monoclonal antibodies in complex with the SARS-CoV-2 spike receptor binding domain, we performed analysis of molecular determinants of antibody recognition of the spike glycoprotein, mapping key residues through analysis of atomic contacts and computational modeling to identify molecular hotspots. Clustering was used to identify four major groups of antibodies based

cov3d.ibbr.umd.edu/download All other relevant data are within the manuscript and its Supporting Information files.

**Funding:** This work was supported by National Institutes of Health grant R01 GM126299 (to BGP). The funders had no role in study design, data collection and analysis, decision to publish, or preparation of the manuscript.

**Competing interests:** The authors have declared that no competing interests exist.

on target residues, and we compared epitope conservation and impact of SARS-CoV-2 variant mutations, showing that certain sets of antibodies predicted to be affected by those variants, while others are capable of targeting escape variants. This analysis can serve as a useful reference for vaccine and immunotherapeutic studies, and we provide updated classifications of antibodies to the research community on our CoV3D site.

## Introduction

Over the past year, the SARS-CoV-2 pandemic has resulted in a massive and growing global death toll and disease burden. A number of vaccines [1], monoclonal antibodies [2], and small molecule therapies [3] that target SARS-CoV-2 have been developed. However, viral variants have raised the possibility of viral escape from, or reduced efficacy of, current vaccines and therapeutics [4–9].

Several recent studies have used in vitro experimental approaches to test human sera [8,10] and sets of monoclonal antibodies [5,8,11,12] to profile SARS-CoV-2 antibody resistance. The rapidly expanding set of experimentally determined structures of antibodies targeting the spike glycoprotein provides the opportunity to use computational biology tools to map key features of antibody-spike recognition. At the same time, the impact of viral variability can be predicted, which can provide insights into effective targeting and neutralization of SARS-CoV-2 and enable selection and engineering of anti-spike therapeutics and vaccines.

Here we report detailed structural analysis of a large set of high resolution antibody-spike complexes that have been collected in our database, CoV3D [13]. Structure-based mapping of antibody footprints on the receptor binding domain (RBD) and unsupervised clustering led to the identification of four major antibody groups based on their recognition signatures. These antibody-spike complexes were assessed for key energetic features using computational alanine mutagenesis of all RBD interface residues to identify shared and distinct binding hotspots on the RBD. The structure-based antibody clusters were also assessed both for residue conservation with SARS-CoV-1, and predicted effects of individual RBD substitutions from circulating SARS-CoV-2 variants, showing substantial differences between groups of RBD-targeting antibodies. These structural features and clusters can serve as a reference for rational vaccine design and therapeutic efforts, and updated antibody cluster information is available to the community on the CoV3D site: https://cov3d.ibbr.umd.edu/antibody_classification.

## Results

### Clustering of antibody-RBD interaction modes

To identify common recognition modes and key features of antibody recognition of the spike glycoprotein, we analyzed a set of high resolution structures of antibody-spike complexes from the CoV3D database [13], which were originally obtained from the Protein Data Bank [14]. We focused on the SARS-CoV-2 RBD, which is the primary target of neutralizing antibodies [15] and is the target of the vast majority of structurally characterized SARS-CoV-2 antibodies. Structures were filtered by resolution ($< 4.0$ Å) and nonredundancy, resulting in 70 antibody-RBD complex structures, representing different antibody formats (heavy-light antibody, nanobody) and a range of IGHV genes (**S1 Table**). As noted in **S1 Table**, all structures were obtained by X-ray diffraction or cryogenic electron microscopy (cryo-EM), and while the cryo-EM structures had significantly lower resolutions ($p < 0.001$), as expected, antibody-RBD interface size and number of inter-molecular atomic contacts were also somewhat lower

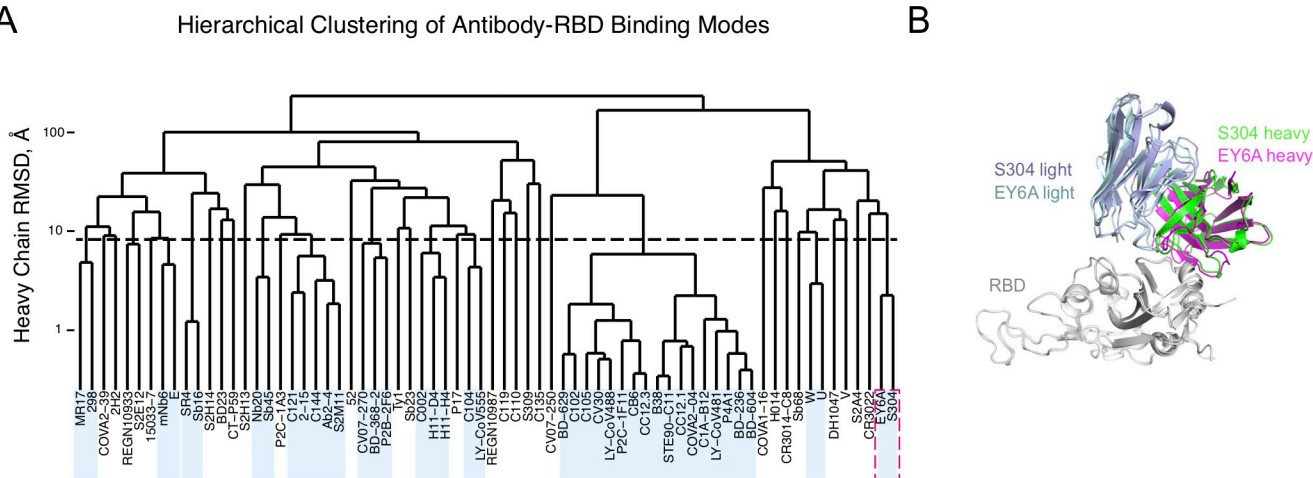

**Fig 1. Hierarchical clustering of SARS-CoV-2 RBD antibody binding modes.** (A) Pairwise root mean square distances (RMSDs) between heavy chain or nanobody binding orientations were determined for 70 antibody-RBD complex structures and used to perform hierarchical clustering. Boxes denote clusters containing multiple antibodies at distance cutoff of 8 Å (shown as dashed horizontal black line), and dashed magenta square denotes co-clustered structures shown in panel (B). (B) Example of co-clustered antibodies S304 (PDB code 7JX3) [21] and EY6A (PDB code 6ZCZ) [22] with a shared RBD binding mode (2.2 Å heavy chain orientation RMSD). Structures are superposed by RBD (gray), and S304 and EY6A heavy and light chains are colored separately as indicated.

for cryo-EM structures, albeit with less significance (**S1 Fig**). The complex structures in this set include multiple therapeutic monoclonal antibodies that have been under clinical investigation: REGN10933 and REGN10987 (casirivimab/imdevimab; REGN-COV2) [16], LY-CoV555 (bamlanivimab) [17], and S309 which is the basis for VIR-7831 (GSK4182136; sotrovimab) [18].

To assess prevalent or shared binding modes in antibody-RBD recognition, pairwise root-mean-square-distances (RMSDs) between antibody heavy chain and nanobody chain orientations were calculated after superposition of RBD coordinates into a common reference frame, and the RMSDs were input to hierarchical clustering analysis (**Fig 1**). This analysis identified a set of 17 complexes with a common binding mode and shared heavy chain germline genes (IGHV3-53, IGHV3-66), a feature that has been noted in previous studies describing SARS-CoV-2 antibody-RBD complex structures [19,20]. Other sets of co-clustered antibodies within the 8 Å RMSD cutoff were limited to antibody pairs, with the exception of a set of five antibodies, of which three (2–15, Ab2-4, C121) share the IGHV1-2 heavy chain germline gene, suggestive of another germline-mediated binding mode. However, other antibodies possessing the IGHV1-2 germline gene exhibited distinct binding modes based on the clustering analysis (298, S2E12), indicating that the heavy chain CDR3 sequence and light chain are relevant factors for that orientation. An example of co-clustered antibodies based on this analysis is shown in **Fig 1B**, showing a shared RBD binding mode (heavy chain orientation RMSD: 2.2 Å) for neutralizing antibodies S304 [21] and EY6A [22], and additional examples of co-clustered pairs are shown in **S2 Fig**.

## High resolution antibody footprinting and clustering analysis

To further delineate features underlying antibody-RBD recognition, we analyzed detailed antibody footprints on the RBD with unsupervised clustering, using the number of atomic contacts by an antibody to each RBD residue as input. Individual antibody footprints and resultant clusters are shown in **Fig 2** (a more detailed heatmap including more RBD residues is

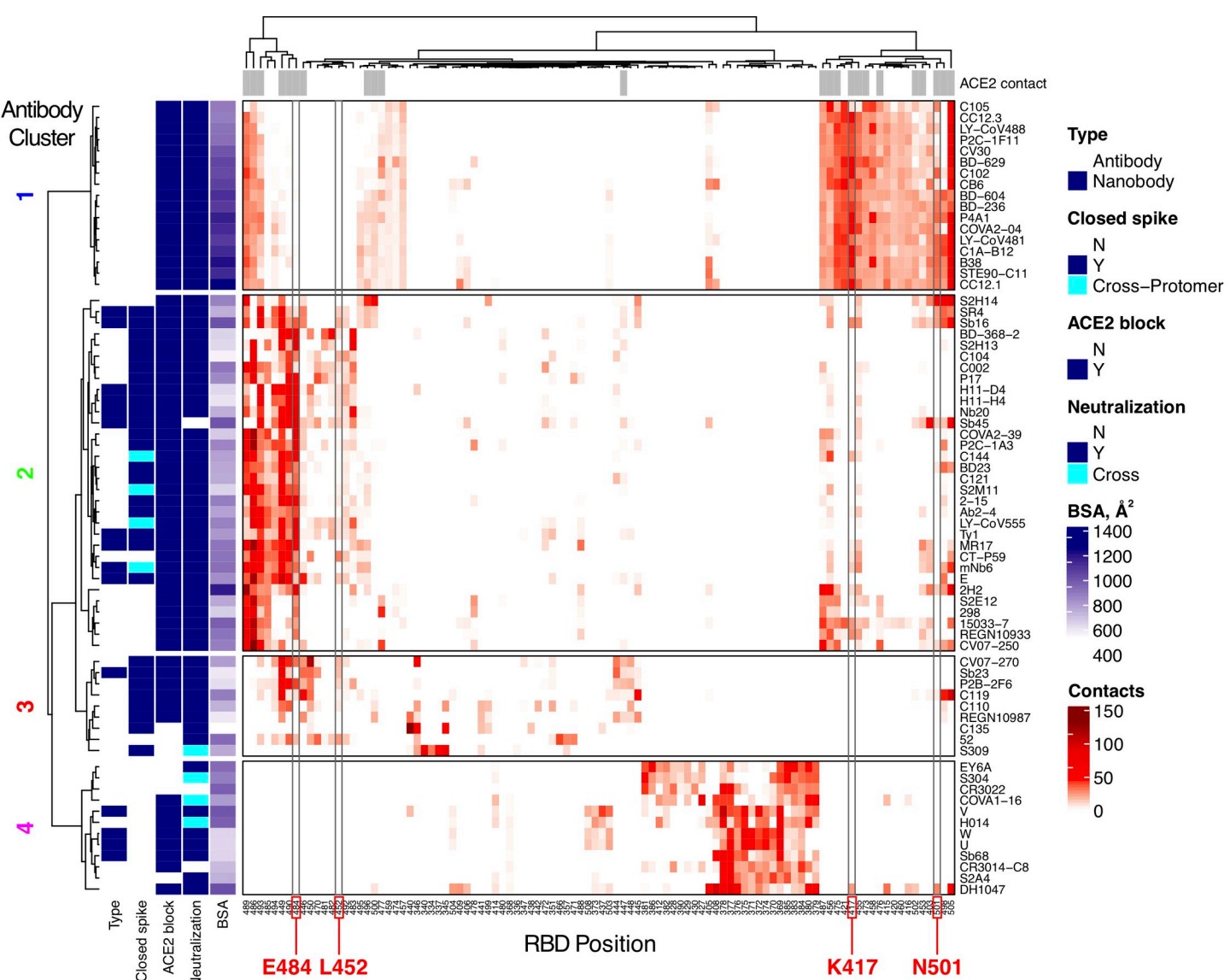

**Fig 2. High resolution mapping and clustering of SARS-CoV-2 RBD antibody binding.** RBD residue contact profiles were generated for each antibody based on number of antibody atomic contacts for each RBD residue within a 5 Å distance cutoff. RBD residues and antibodies are ordered using hierarchical clustering analysis, with dendrograms shown on top and left. The antibodies are separated into four major clusters based on contact profiles, and cluster numbers (1–4) are indicated on left. Contacts in heatmap are colored by number of RBD residue antibody atomic contacts, as indicated in the key. For reference, antibody type (Antibody: heavy-chain antibody, Nanobody: single-chain antibody), binding to RBD-closed spike conformation (Closed spike), ability to block ACE2 binding (ACE2 block), SARS-CoV-2 neutralization or SARS-CoV-2/SARS-CoV-1 cross-neutralization ("Y" and "Cross", respectively, under Neutralization), and interface buried surface area (BSA, Å²) are shown on the left sidebars. Closed spike binding and ACE2 blocking were calculated based on the structures, as described in the Methods. The top bar above the heatmap indicates RBD residues contacted by ACE2 (5 Å distance cutoff) in an ACE2-RBD complex structure (PDB code 6LZG) [52]. For clarity, 100 RBD residues are shown in heatmap; a heatmap with the full set of 139 contacted RBD residues which was used to cluster the antibodies in this figure is shown in S2 Fig. RBD residues that are mutated in SARS-CoV-2 variants of concern (K417, L452, E484, N501) are labeled at bottom and highlighted with gray boxes in heatmap.

given in S3 Fig) along with calculated and previously reported properties of the antibodies for reference, including interface buried surface area (BSA), neutralization (SARS-CoV-2 neutralization or SARS-CoV-1/SARS-CoV-2 cross-neutralization), ACE2 blocking, and capability to bind the RBD in the context of the closed (or down) spike conformation. This separated the antibodies into four main clusters; these are similar but not identical to previously described SARS-CoV-2 antibody classifications [23], which are shown as the "BBclass" colored sidebar

in **S3 Fig**. Inspection of the heatmap indicates that Clusters 1 and 4 are most distinct, which is supported by high bootstrap confidence levels (100% and 99% respectively, **S4 Fig**), while Clusters 2 and 3 are more diverse, and have bootstrap confidence levels of 87% and 83% (**S4 Fig**). Due to the moderately lower bootstrap confidence, it is possible that some antibodies from Clusters 2 and 3, particularly those proximal to the inter-cluster boundaries and including some cryo-EM structures that have poorer resolutions (**S1 Fig**), could have potential ambiguity in Cluster 2 versus Cluster 3 assignments. Visualization of the distribution of the antibody positions on the RBD surface (**Fig 3**) shows that Clusters 1 and 2 are spatially proximal and overlap with the ACE2 binding site, and the relatively constrained positions of Cluster 1 antibodies are reflective of our RMSD-based analysis (**Fig 1**) and known conserved binding mode of that set. Cluster 3 extends to the RBD hinge and N-glycan at RBD position N343, while Cluster 4 occupies a distinct region of the RBD. Principal component analysis using the antibody atom contact data as input enabled visualization of the antibody distributions along the first two principal components, which collectively represent approximately 50% of the data (**S5 Fig**), and generally supports the hierarchical clustering.

The contact-based clusters in **Fig 2** highlight several notable features within and between sets of RBD-targeting antibodies. Cluster 1 antibodies all neutralize SARS-CoV-2, block ACE2 binding, can only bind the spike in its open conformation, and have relatively high RBD interface buried surface area (BSA). Cluster 2 contains antibodies that can bind the closed spike, some of which can engage multiple RBDs in that context, and all are predicted or confirmed to block ACE2 binding. Cluster 3 is dominated by antibodies that can bind the closed spike, and most Cluster 3 antibodies are predicted to block ACE2 binding through steric hindrance and/ or binding site overlap. In Cluster 4, which is mapped closer to the N- and C-termini and the hinge that connects the RBD to the spike (**Fig 3**), multiple antibodies are confirmed to be cross-neutralizing between SARS-CoV-2 and SARS-CoV-1 [21,24,25], and no antibodies are predicted to recognize spike in the RBD-closed conformation. The mapped antibody footprints show varying degrees of overlap with ACE2 binding site residues (gray bars at top of **Fig 2**) among the clusters. Residues highlighted in **Fig 2** that are associated with viral variants of concern (E484, L452, K417, N501) show that Cluster 2 is primarily associated with E484 engagement, Cluster 1 is associated with engagement of K417 and N501, while residue L452, which is mutated in the Delta variant, contacts many of the antibodies in Clusters 2 and 3. Antibodies in Cluster 4 exhibit few or no contacts with those residues, suggesting that they are less susceptible to binding disruption and viral resistance due to variability at those sites.

## Binding energetic features and hotspots

To provide a more detailed and comprehensive view of key residues and energetic features underlying antibody-RBD recognition, all interface structures were analyzed for hydrogen bonds with RBD residues (**Fig 4**) and energetically important RBD residues based on computational alanine scanning (**Fig 5**). Hydrogen bonding patterns in RBD-targeting antibodies (**Fig 4**) showed clear preferences for hydrogen bond RBD residue interactions among Cluster 1 antibodies, with frequently observed interactions with residues R403, K417, D420, Y421, N487, and Y505. Many Cluster 2 antibodies exhibit hydrogen bond interactions with residue E484 and/or Q493, whereas antibodies from Clusters 3 and 4 have limited shared RBD residues involved in hydrogen bond interactions.

To map key RBD sites and energetic hotspots in the set of antibody-RBD interfaces, we performed computational alanine scanning (**Fig 5**) using a mutagenesis protocol in Rosetta [26]. The protocol used for this analysis was selected based on predictive performance from benchmarking of nine computational methods using approximately 350 experimentally determined

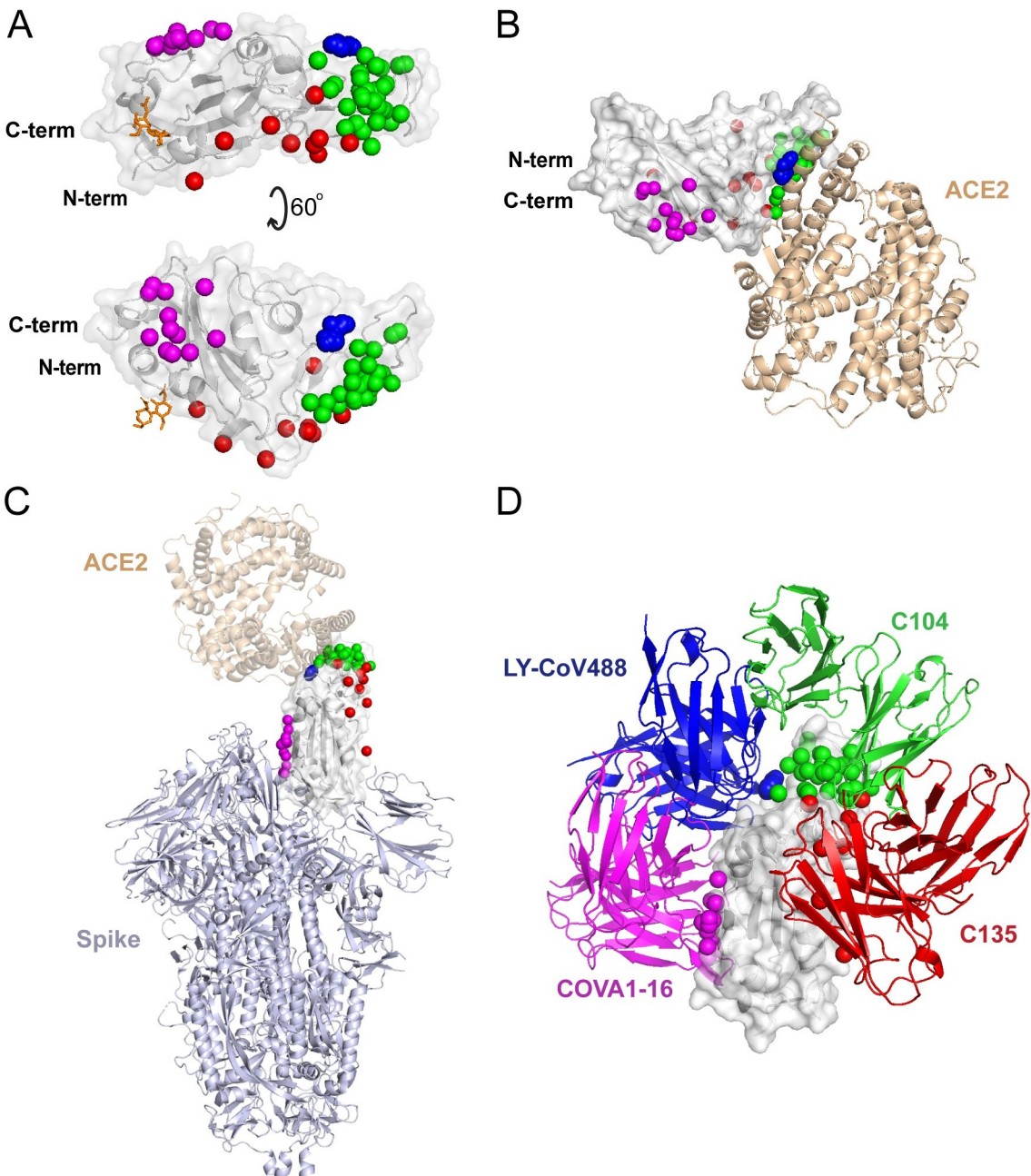

**Fig 3. Distribution of antibody clusters on the receptor binding domain.** (A) Each antibody is represented as a sphere at the paratope center (centroid of all non-hydrogen atoms within 5 Å of the RBD), and colored by contact-based antibody cluster (1: blue, 2: green, 3: red, 4: magenta). A representative RBD structure (from PDB code 7KN5) is shown in gray, and the N-glycan at residue N343 from that structure is shown as orange sticks. (B) RBD structure with antibody clusters and superposed ACE2 receptor (tan cartoon; PDB code 6LZG [52]). (C) RBD antibody clusters shown in the context of the spike glycoprotein (light blue cartoon; PDB code 6VYB [68]) with the RBD in an open state. (D) Representative antibodies from each cluster, labeled by antibody name and colored by cluster, superposed onto the RBD.

alanine mutant ΔΔG values for antibody-antigen interfaces (**S2 Table**). While many energetically important residues identified by this analysis are reflective of the key residues identified by hydrogen bond analysis, including residues N487 and E484 (Cluster 1) and E484 (Cluster 2), numerous hydrophobic RBD residues were additionally identified as important for binding

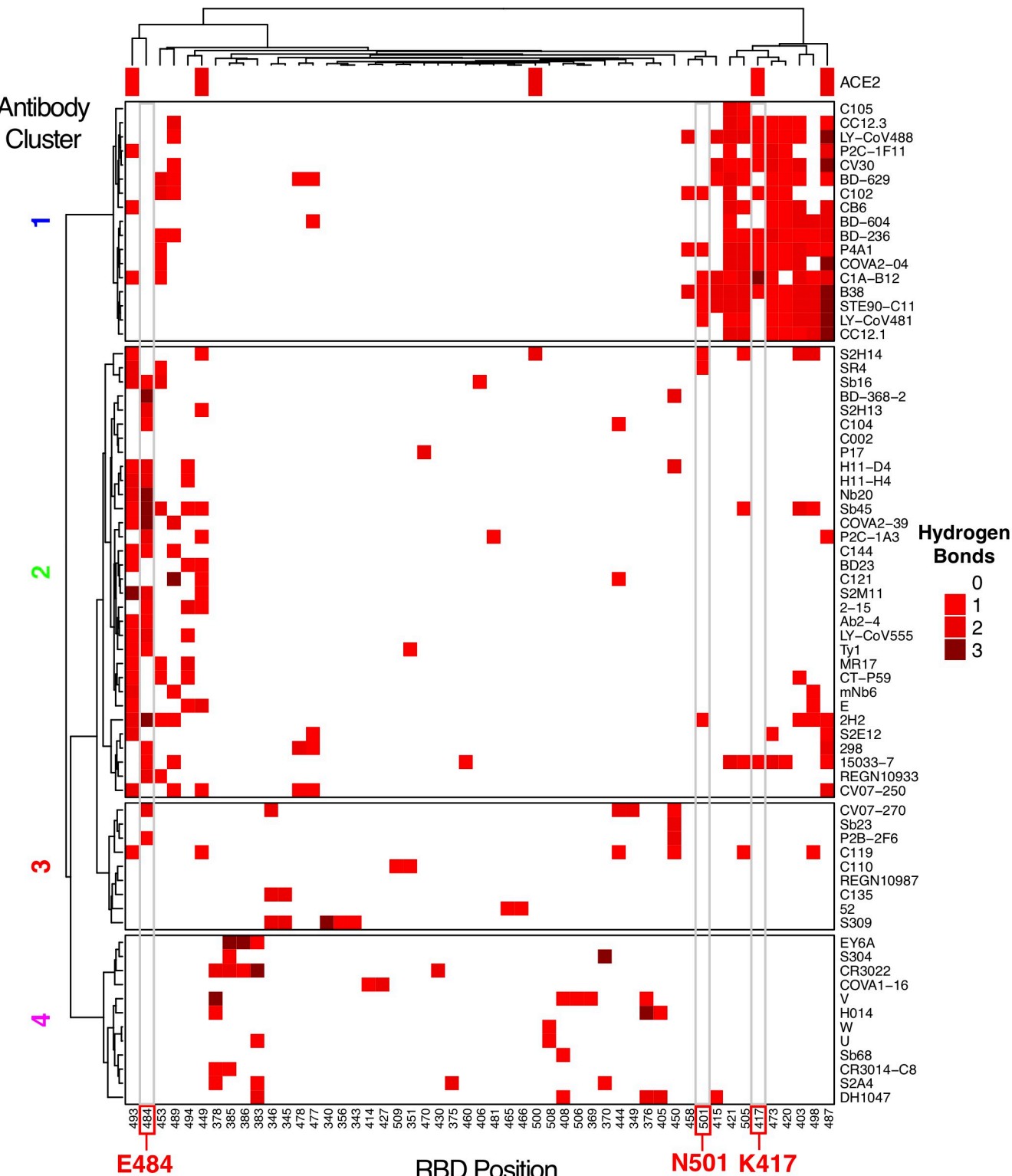

**Fig 4. RBD hydrogen bond contacts of SARS-CoV-2 antibodies.** Hydrogen bonds to RBD residue side chains were calculated for all antibody-RBD complexes using the hbplus program [51]. Each hydrogen bond contact is colored by number of hydrogen bond interactions, as indicated on the key, and RBD positions are ordered by hierarchical clustering based on hydrogen bond profile similarities, with corresponding dendrogram shown at top. Antibodies (rows) are ordered and clustered as in **Fig 2**, based on the RBD contact profile similarities, and RBD hydrogen bond contacts with ACE2 (PDB code 6LZG) are shown in the top bar. RBD residues that are mutated in SARS-CoV-2 variants of concern (K417, E484, N501) are labeled at bottom and highlighted with gray boxes in heatmap.

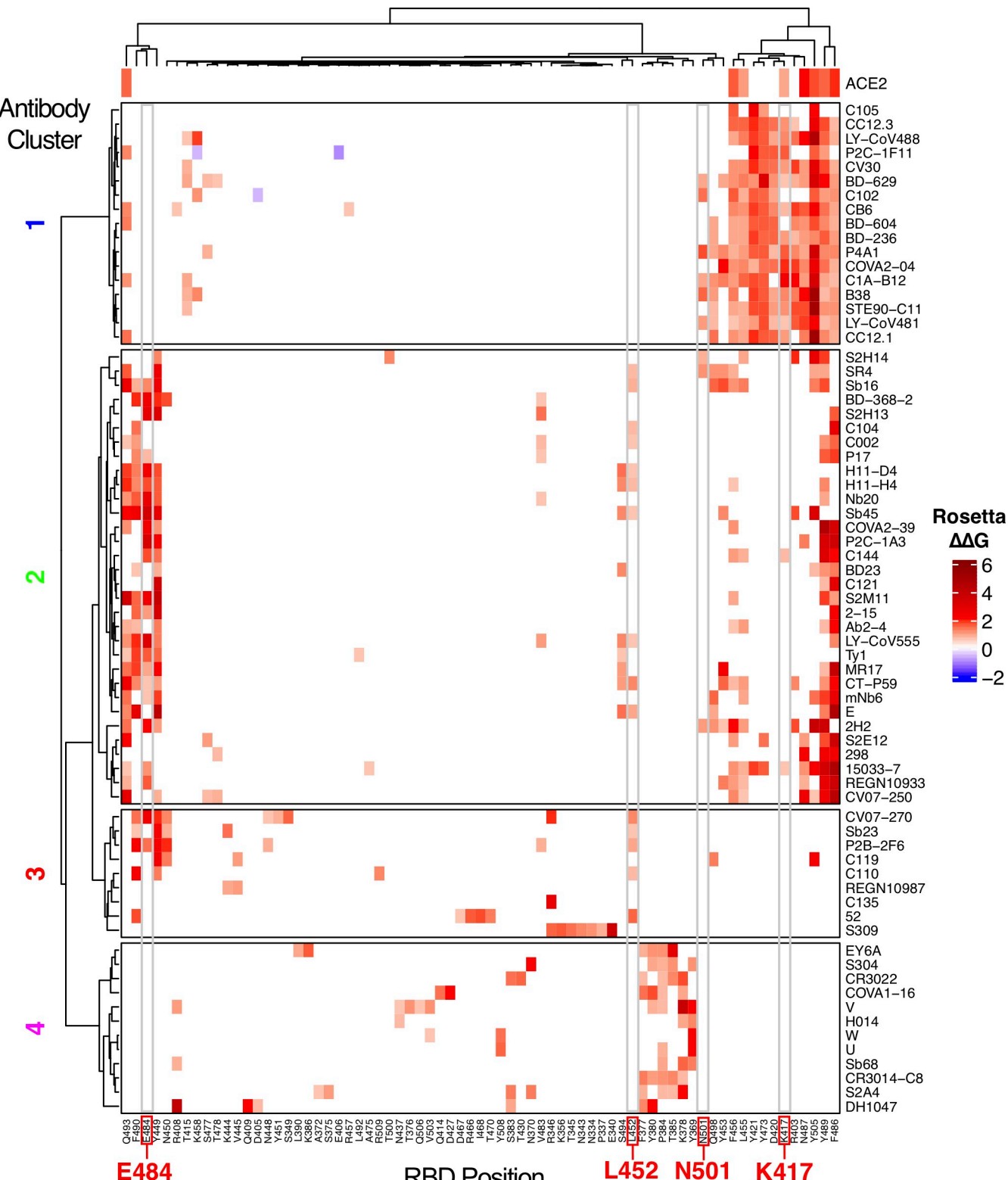

**Fig 5. Computational mapping of SARS-CoV-2 RBD hotspot residues.** Computational alanine scanning of RBD residues in antibody-RBD interfaces was performed using Rosetta [26], to generate binding energy change (ΔΔG) values for alanine substitutions at each RBD position based on modeling of residue substitutions and scoring using an energy-based function. ΔΔG values are in Rosetta Energy Units (REU) which are comparable to energies in kcal/mol. Alanine residues in the native complex were mutated to glycine for ΔΔG calculations, and glycine RBD residues were omitted from the analysis. In order to

highlight substantial predicted binding energy changes, only ΔΔGs with absolute values > 0.5 REU are represented. RBD residues are ordered by hierarchical clustering based on ΔΔG profile similarities, with corresponding dendrogram shown at top. Antibodies (rows) are ordered and clustered as in **Fig 2**, based on the RBD contact profile similarities. For reference, ΔΔGs for ACE2 binding based on the ACE2-RBD complex structure (PDB code 6LZG) are shown in the top bar. RBD residues that are mutated in SARS-CoV-2 variants of concern (K417, L452, E484, N501) are labeled at bottom and highlighted with gray boxes in heatmap.

within antibody clusters. These residues include Y505 (Cluster 1), F486 and Y489 (Clusters 1 and 2), and Y449 and F490 (Clusters 2 and 3). As with the analysis of RBD residue contacts, analysis of hydrogen bonds and computational alanine scanning support the overall importance of N417 and Y501 for Cluster 1 antibodies, and E484 for Cluster 2 antibodies. While residue L452 is present in **Fig 2**, it is not present in **Fig 5** as the hydrophobic leucine residue does not form antibody hydrogen bonds.

## Epitope conservation and targeting of escape variants

To assess the degree to which antibodies of different classes can target sites that are conserved among sarbecoviruses, we calculated the fraction of RBD epitope residues conserved between SARS-CoV-2 and SARS-CoV-1 for each antibody-RBD interface (**Fig 6**). Antibodies in Clusters 1–3 exhibit limited conservation (approximately 50% or lower conserved antibody contact residues), with the exception of S309, which shows over 80% epitope residue conservation; this result is in accordance with the observed cross-neutralizing capability for that antibody [27]. In contrast with the other antibody clusters, antibodies in Cluster 4, which includes three confirmed cross-neutralizing antibodies (**Fig 2**), exhibit markedly higher epitope conservation,

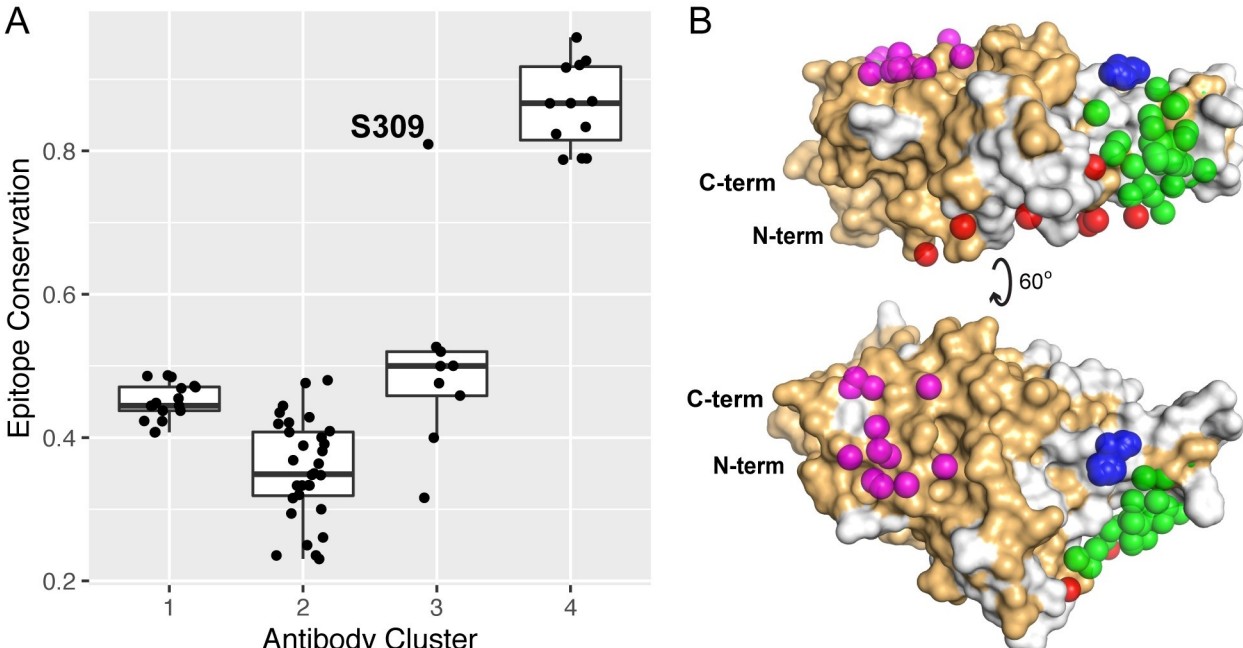

**Fig 6. Epitope residue conservation in SARS-CoV-1 by antibody cluster.** (A) Epitope conservation, defined as the fraction of RBD epitope residues (< 5 Å distance to antibody) conserved between SARS-COV-1 and SARS-COV-2, was calculated for 70 antibody-RBD complex structures, and conservation values are shown as a boxplot grouped by antibody clusters, with all conservation values shown as points. The outlier point for Cluster 3 (S304 antibody) is labeled, and the total numbers of points are 17 (Cluster 1), 32 (Cluster 2), 9 (Cluster 3), and 12 (Cluster 4). (B) Conserved RBD residues are highlighted on the RBD structure, with conserved RBD residues shown as orange and non-conserved residues gray, and represented as in **Fig 3A** with antibody cluster paratopes as spheres.

with all values 78% or higher. This suggests that the highly conserved site targeted by Cluster 4 antibodies, which is inaccessible in the closed spike conformation, is potentially important in conferring immunity across sarbecoviruses.

To directly assess the effects of RBD mutations present in recently described SARS-CoV-2 variants of concern, we performed computational mutagenesis to gauge whether antibody binding affinities are predicted to be disrupted by individual RBD substitutions, as well as effects on ACE2 binding. For initial simulations, we utilized the same protocol that was used for computational alanine scanning; we found this method to have similar predictive performance for point residue substitutions to all residue types in comparison with performance for alanine-only substitutions (Pearson Correlation Coefficient (PCC) with experimental ΔΔGs of 0.5 for all residues, versus 0.53 for alanine-only; **S2 Table**). RBD substitutions K417N, K417T, L452R, S477N, T478K, E484K, E484Q, and N501Y were modeled in all interfaces and assessed for antibody and ACE2 ΔΔGs; these substitutions are collectively represented in variants of concern Alpha (B.1.1.7; N501Y), Beta (B.1.351; K417N, E484K, N501Y), Gamma (P.1; K417T, E484K, N501Y), and Delta (B.1.617.2; L452R, T478K), and variant of interest Kappa (B.1.617.1; L452R, E484Q). Comparison of predicted ΔΔGs (**Fig 7**) indicates that K417N, K417T, and to a lesser extent N501Y, are predicted to predominantly affect antibodies in Cluster 1, whereas disruptive effects of E484K and E484Q are primarily observed for antibody Cluster 2. Cluster 3 antibodies with predicted ΔΔG values of over 1 Rosetta Energy Unit (REU) were observed for E484 substitutions, but were very limited (two antibodies for E484K, one antibody for E484Q). In contrast, antibodies in Cluster 3 and 4 exhibit little overall predicted effects from the variant RBD point substitutions considered here, and other variants substitutions did not show marked predicted effects on antibody binding. The binding affinity for ACE2 was predicted to decrease for substitutions K417N and K417T, and increase for N501Y, while remaining the same for other substitutions. This is in accordance with recently reported ACE2-RBD binding measurements, where N501Y led to a 2-fold improvement in ACE2 binding, K417N led to a 7-fold loss in ACE2 binding, and E484K maintained ACE2 binding (< 2-fold affinity change) [28]. We also tested predicted binding effects using a different modeling tool (FoldX), which uses a distinct modeling and scoring protocol from Rosetta, and found similar trends among antibody classes for the effects of the variants (**S6 Fig**). However, there are some differences between FoldX and Rosetta ΔΔG predictions, such as the L452R RBD variant, for which FoldX predicted more antibody binding disruptions than Rosetta.

To more directly assess the effects of SARS-CoV-2 variants on antibody binding, we calculated ΔΔGs for combinations of RBD substitutions found in variants of concern Beta, Gamma, and Delta (**Fig 8**). Binding effects for Alpha, which is equivalent to N501Y in **Fig 7** as it contains the same RBD substitution, are also shown in **Fig 8** for reference. Based on comparison of ΔΔG predictions with recently published experimentally measured neutralization results for variants and monoclonal antibodies overlapping with the set in this study (**S3 and S4 Tables**), FoldX was included along with Rosetta in **Fig 8**, as the former showed a modest improvement in sensitivity over the Rosetta ΔΔG protocol, detecting one more antibody-variant pair with loss of neutralization in each of **S3 and S4 Tables**. Overall, the comparison of measured neutralization changes and predicted ΔΔG values shows that the structure-based affinity predictions can in most cases reflect neutralization effects. Predicted ΔΔGs for the antibody clusters from Rosetta (**Fig 8A**) indicated that the Alpha, Beta, and Gamma variants are disruptive for Cluster 1 antibodies, and Beta and Gamma variants are disruptive for Cluster 2 antibodies; those results are generally in agreement with FoldX (**Fig 8B**). However, in contrast with Rosetta which predicted minor effects from the Delta variant on antibody recognition, FoldX predicted that the Delta variant would markedly disrupt antibody binding in Clusters 2 and 3. Given its modestly higher performance in the comparison with experimentally determined

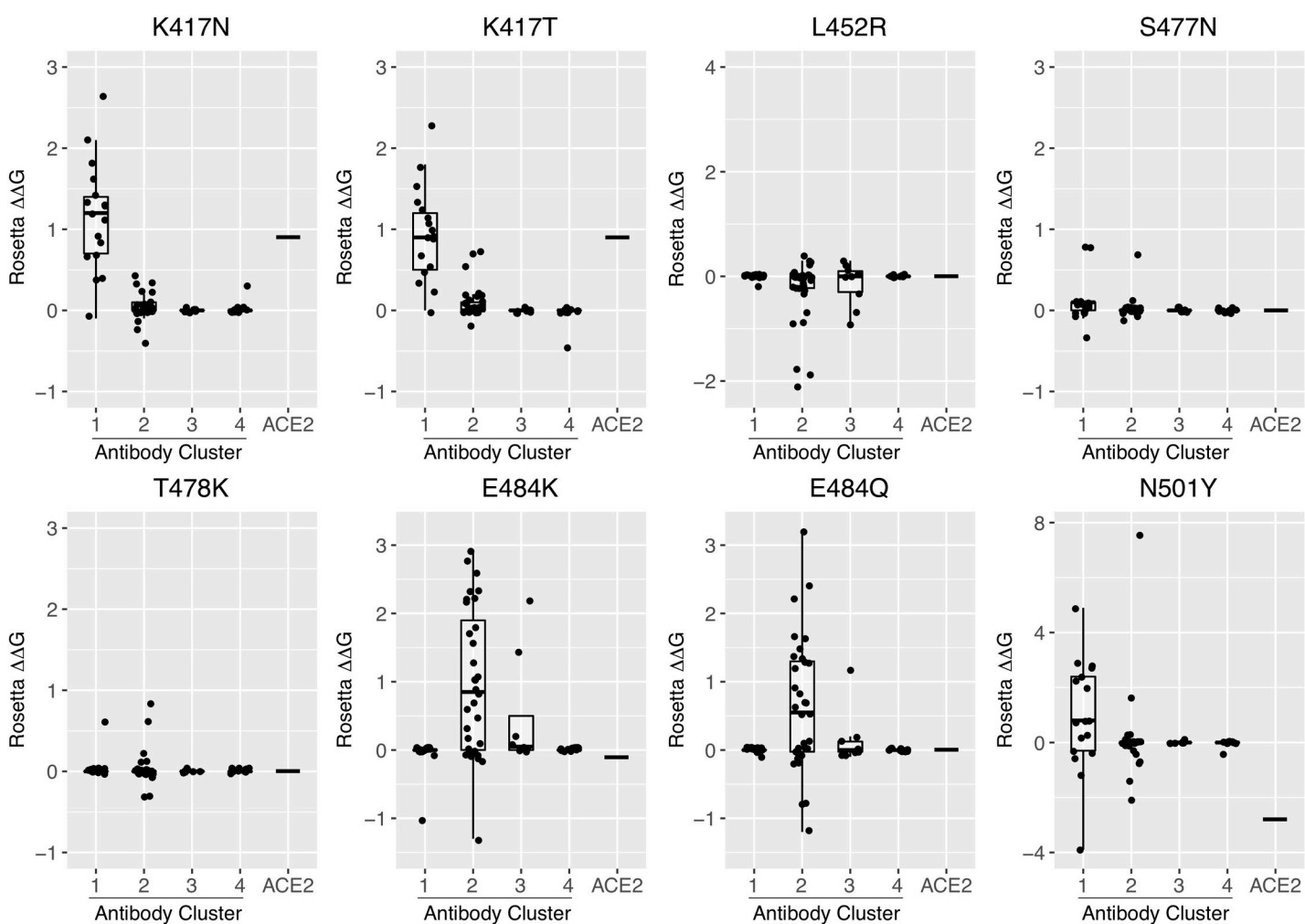

**Fig 7. Profiling antibody and receptor binding effects of RBD point substitutions from circulating SARS-CoV-2 variants.** Computational mutagenesis in Rosetta [26] was used to predict binding affinity effects (ΔΔGs) of RBD variant substitutions K417N, K417T, L452R, S477N, T478K, E484K, E484Q, and N501Y for 70 antibodies that target the RBD, as well as the ACE2 receptor. ΔΔG values are shown as boxplots grouped by antibody clusters, with all antibody ΔΔG values shown as points, and the ACE2 ΔΔG value represented as a horizontal bar in each boxplot. ΔΔG values are in Rosetta Energy Units (REU), which are comparable to energies in kcal/mol.

variant neutralization effects (**S3 and S4 Tables**), the predictions of disruption from the FoldX protocol seem more likely to reflect the antibody binding effects for that variant.

## Discussion

Utilizing a curated set of experimentally determined antibody-RBD complex structures, we have performed detailed mapping of antibody recognition determinants on the SARS-CoV-2 RBD, which were used to identify antibody clusters that exhibit distinct structural and energetic signatures. Notably, these clusters exhibited different destabilizing effects for RBD substitutions found in circulating variants, expanding upon previous observations by others on the effects of specific substitutions such as E484K for specific groups of antibodies [23,28,29]. We found that Cluster 2 antibodies, which overlap with Class 2 antibodies reported by Barnes et al. [23], are susceptible to resistance from SARS-CoV-2 variants with the E484K substitution, which include Beta (B.1.351) and Gamma (P.1), whereas other antibodies are not likely to be

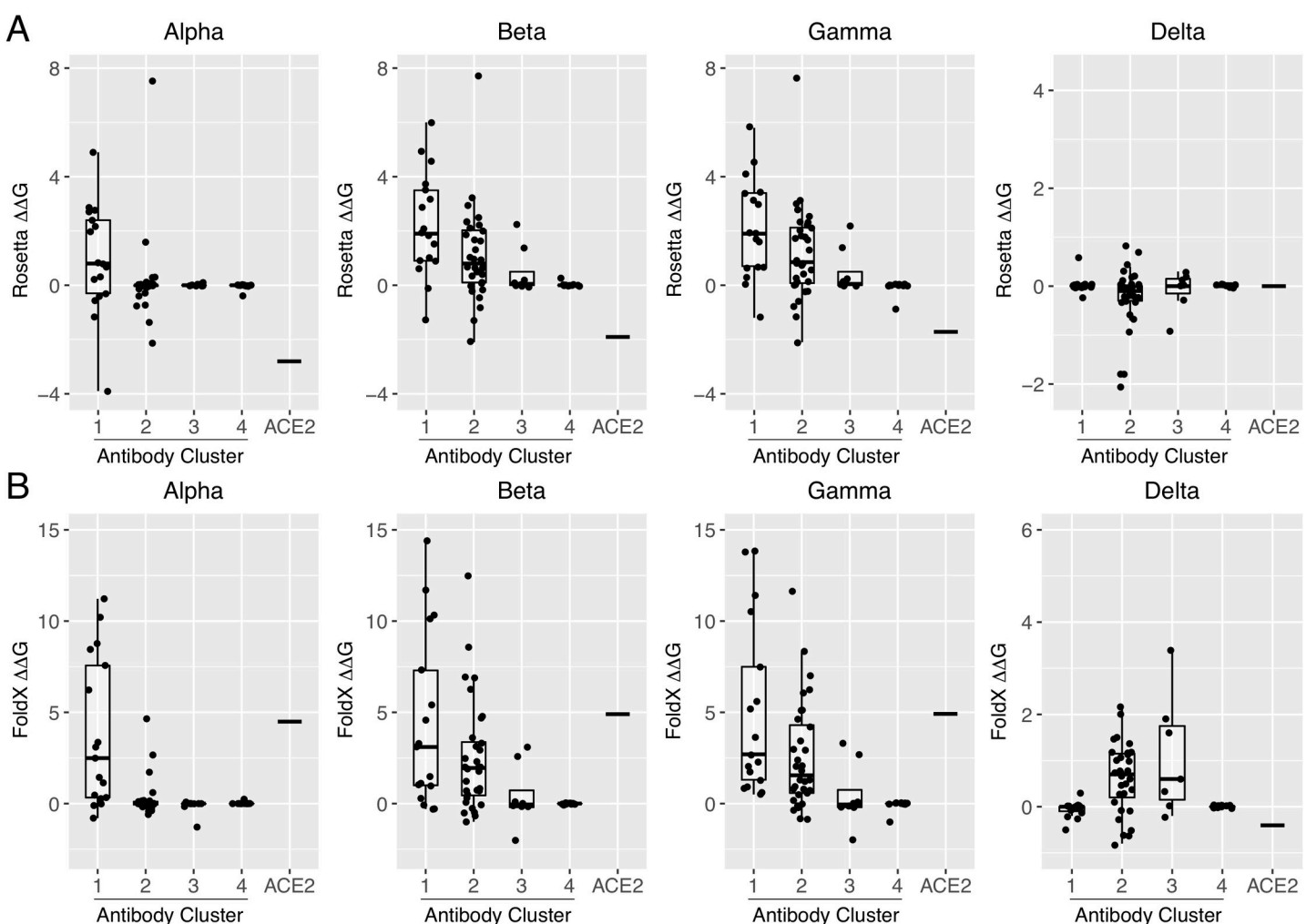

**Fig 8. Profiling antibody and receptor RBD binding effects for circulating SARS-CoV-2 variants.** Computational mutagenesis was used to predict binding affinity effects (ΔΔGs) of SARS-CoV-2 variants of concern Alpha (B.1.1.7; RBD substitution N501Y), Beta (B.1.351; RBD substitutions K417N, E484K, N501Y), Gamma (P.1; RBD substitutions K417T, E484K, N501Y), and Delta (B.1.617.2; RBD substitutions L452R, T478K), using (A) Rosetta and (B) FoldX. ΔΔG values are shown as boxplots grouped by antibody clusters or ACE2 receptor, with all antibody ΔΔG values shown as points, and the ACE2 ΔΔG value represented as a horizontal bar in each boxplot. Both Rosetta and FoldX ΔΔG values are commensurate with energies in kcal/mol.

affected by that substitution. In contrast, substitutions at residues K417 and N501, which are found in several variants of concern, were primarily associated with binding disruption for Cluster 1 antibodies based on our computational mutagenesis. Given that the E484K substitution appears specifically associated with viral escape, as noted by others [30] and supported by recent studies of monoclonal and polyclonal antibody neutralization of variant viruses and specific mutants [7,8], our work highlights the relative importance of Cluster 2 antibodies in the neutralizing response against SARS-CoV-2 due to natural infection or immunization.

Our analysis highlights the ability of computational structure-based protocols to rapidly predict and profile resistance for new and emerging SARS-CoV-2 variants. This is exemplified by our results for the Delta variant, which was designated a variant of concern (VOC) in May 2021 and is responsible for a recent global rise in COVID-19 cases. We found that the Delta variant is predicted to be resistant to antibodies in Clusters 2 and 3, and this is likely driven by the L452R RBD substitution. This resistance is corroborated by recent reports of monoclonal

antibody resistance [31] and lower neutralization in vaccinated individuals [32] for the Delta variant, which can lead to breakthrough infections in some cases [33]. Based on our predictions for the effects of K417N (**Figs 7 and S6**), the "Delta plus" variant which includes that mutation would likely exhibit resistance to additional antibodies, including antibodies in Cluster 1, albeit with possibly reduced ACE2 binding.

This study is distinguished from other recently described structure-based [23] and binding competition-based [21,34] reports to compare and classify antibodies that target SARS-CoV-2, as we directly assessed detailed antibody binding signatures, generated using atomic contact counts to RBD residues, and used unsupervised clustering with these features to generate the resultant classes. Furthermore, we generated an energetic map of RBD antigenicity based on comprehensive computational alanine scanning mutagenesis. To provide an updated reference to the community, we report these clusters on our CoV3D site of coronavirus protein structures [13] (https://cov3d.ibbr.umd.edu/antibody_classification), which includes the 70 complexes reported in this study as well as newly reported complexes. We also provide a prototype interface on the CoV3D site for the community to input new experimentally determined structures or models of antibody-RBD or protein-RBD complexes to characterize binding footprints and assign contact-based clusters.

Certain elements of our analysis of antibody binding determinants can be expanded in future studies. The calculation of antibody contacts and energetic determinants on the RBD did not include non-protein atoms, such as water molecules and N-glycans, and in some cases, certain residues were disordered in the experimentally determined structures. Water molecules, which could mediate hydrogen bonds between antibody and RBD, were not included here, to avoid bias due to varying experimental structural resolutions which in many cases could not resolve water molecules, necessitating modeling of explicit water molecules which would lead to additional uncertainties in subsequent calculations [35]. Likewise, the N-glycans of the RBD, specifically the glycan at residue N343, has varying occupancies in experimentally determined structures. Though this glycan is contacted by the S309 antibody [27], such glycan contacts appear to be rare in antibody-RBD complex structures, at least for structurally characterized neutralizing antibodies, of which most compete with ACE2 binding and thus target sites that are not proximal to that N-glycan. Modeling of missing N-glycans, water molecules, and any missing residues may still provide possible insights into recognition features, as well as simulations of interface molecular dynamics, or docking simulations of separated antibody and RBD molecules to assess binding energy funnels [36]. Predictive computational docking and template-based modeling can also be used to generate antibody-RBD complex models for antibodies with sequences available but no known structure, enabled in part by databases containing sequences of RBD-targeting antibodies [37]. An additional avenue for expansion would be the analysis of antibodies that target other regions of the spike glycoprotein, including the N-terminal domain (NTD), which have been described in recent structural and antigenic mapping studies [38,39]. We currently represent this set as the "NTD" antibody class on the CoV3D site, and may perform a more detailed energetic and footprinting analysis of this set in the future.

In addition to providing a view of the detailed landscape of antibody-RBD recognition determinants and key sites, our results indicate that certain sets of antibodies are less susceptible to resistance from variants and have higher average epitope sequence conservation with SARS-CoV-1. Furthermore, several of the antibodies in Cluster 4 have been experimentally confirmed cross-neutralize SARS-CoV-1 and SARS-CoV-2. Recently reported broadly reactive RBD-binding antibodies that recognize human and zoonotic SARS-like coronaviruses (sarbecoviruses) [40–42] can provide additional structural data to map these key conserved regions and epitopes. Prospective structure-based antigen design studies could potentially focus the

antibody response to the corresponding epitopes of the SARS-CoV-2 RBD, versus the epitopes collectively targeted by antibodies in Clusters 1 and 2. As binding of Cluster 4 antibodies is prevented in the context of the closed-RBD spike conformation, open spike antigen designs or RBD-only antigens would likely facilitate elicitation of these antibodies. Several recent studies have reported success using RBDs displayed on self-assembling nanoparticles [43–45], and structure-guided RBD optimization in the context of such a platform could lead to improved elicitation of antibodies associated with a cross-sarbecovirus response. Such antigen design efforts could result in an effective vaccine that provides protection against SARS-CoV-2 variants as well as future emerging coronaviruses.

## Materials and methods

### Structure assembly and curation

Structures of antibody-RBD complexes were downloaded from the CoV3D database [13], which identifies antibody-RBD structures in the Protein Data Bank [14] on a weekly basis through sequence similarity to coronavirus reference protein sequences in conjunction with identification and annotation of antibody chains. The set of antibody-RBD structures (downloaded in February 2021) was filtered for antibody nonredundancy based on antibody name and sequence identity, as well as resolution ($< 4.0$ Å). In cases of an antibody present in multiple antibody-RBD complex structures, the structure with highest resolution was selected for analysis. For consistency among antibody-RBD complex structures, and to facilitate calculations, antibodies were truncated to include variable domains, and full spike glycoproteins were truncated to include only RBD residues (residues 333–527) of the sole or primary target of the antibody. To provide uniform input structures for atomic contact and other calculations, non-amino acid HETATMs were removed prior to structural analysis, and to resolve double occupancies and add missing side chain atoms, structures were pre-processed by the "score" application in Rosetta version 3.12 [46]. Two complexes with missing side chain atoms in the experimental PDB coordinates were processed using the FastRelax protocol in Rosetta [47], to perform constrained local minimization and to resolve unfavorable energies due to clashes from rebuilt side chains (antibodies DH1047, C104; PDB codes 7LD1, 7K8U). Parameter flags used in FastRelax ("relax" executable in Rosetta 3.12) are:

```
-relax:constrain_relax_to_start_coords
-relax:coord_constrain_sidechains
-relax:ramp_constraints false
-ex1
-ex2aro
-no_optH false
-flip_HNQ
-renumber_pdb F
-nstruct 1
```

The set of pre-processed structures, aligned to a common RBD reference frame, is available through the CoV3D site [13], at: https://cov3d.ibbr.umd.edu/download ("Nonredundant RBD-antibody complex structures" link).

Information regarding neutralization of SARS-CoV-2 and SARS-CoV-1 was obtained from the CoV-AbDab site [37], as well as from the literature for certain antibodies, where noted in **S1 Table**.

### Computational structural analysis

RMSD values between antibody heavy chain or nanobody orientations were determined by superposition of RBDs from two complexes using least-squares fitting of backbone atoms,

followed by superposition of one antibody variable domain onto another using least squares fitting of framework residue backbone atoms, and calculation of backbone RMSD between superposed and non-superposed variable domain. RBD residues used for superposition (present in all structures in this set) are 338–356, 375–382, 397–442, 448–454, 462–467, 490–501, and 503–514. Antibody variable domain framework residues used for superposition and RMSD calculations are 3–7, 21–24, 41–46, 52–57, 78–82, 89–93, 102–108, and 141–144, based on the AHo numbering system [48]. Interface contacts are defined as inter-atomic distance between non-hydrogen atoms of less than 5 Å, and antibody-RBD residue contact maps were generated based on the total number of antibody atom contacts with each RBD residue. Hierarchical clustering of antibody RMSDs was performed in R version 4.0.3 (www.r-project.org) with the distance matrix of RMSDs as input, and Ward's minimum variance method ("ward. D2" method in hclust). Hierarchical clustering of antibodies and RBD positions based on contact data was performed in R, using Manhattan distance to compute differences in contact profiles between antibodies or RBD positions, and Ward's minimum variance method for clustering. Hierarchical clustering of RBD positions based on hydrogen bond or calculated ΔΔG values, for the respective heatmap figures, was likewise performed in R, using Manhattan distances and Ward's clustering algorithm. RBD residue dimension reduction for representation in main heatmap (**Fig 2**) was performed by selecting exemplar residues from 100 hierarchical clusters, which removed residues with highly similar contact profiles (based on Manhattan distance) with respect to those shown in the heatmap. The pvclust method [49], as implemented in R, was used to calculate bootstrap confidence of contact-based hierarchical clusters of antibodies, using 20,000 bootstrap replicates. Principal component analysis of antibody-RBD contact profile data was performed with the scikit-learn Python module.

Buried surface areas (BSAs) were calculated using the naccess program (v. 2.1.1) [50], subtracting the solvent accessible surface area of the antibody-RBD complex structure from the total solvent accessible surface area of the separate antibody and RBD structures, dividing by two to avoid double-counting interface area and to make BSA values commensurate with those from other tools including PISA (http://www.ebi.ac.uk/pdbe/prot_int/pistart.html). Antibody-RBD interface hydrogen bonds were calculated using the hbplus program (v. 3.15) [51], with default parameters.

An X-ray structure of the ACE2-RBD complex (PDB code 6LZG) [52] was used to calculate ACE2-RBD residue contacts, hydrogen bonds, ΔΔGs, as well as antibody blocking of ACE2 binding to the RBD. For calculations of ACE2 blocking, after superposition of ACE2-RBD and antibody-RBD complexes by RBD, the number of inter-atomic clashes, defined as non-hydrogen atom pairs with distances < 2.5 Å, was calculated between ACE2 and each antibody structure. Antibodies with > 20 atomic clashes with ACE2 were classified as likely to block ACE2 binding.

Structure-based calculations of antibody binding to the closed spike structure were performed using the SARS-CoV-2 closed spike structure reported by Walls et al. (PDB code 6VXX) [53]. Antibodies with < 100 atomic clashes with spike atoms outside of the target RBD structure and chain after superposition of the antibody-RBD complex onto the 6VXX structure were classified as predicted to bind the closed spike. Clash thresholds were selected based on agreement with structures and experimental data regarding ACE2 blocking and closed spike binding, when available. Four antibodies that engaged the closed spike and exhibited cross-protomer binding, as confirmed by inspection of antibody-spike complex structures (S2M11, C144, mNb6, LY-CoV555; PDB codes 7K43, 7K90, 7KKL, 7L3N) [23,54–56], were annotated accordingly in the contact heatmap.

## Computational mutagenesis

Computational modeling and prediction of antibody binding energy changes (ΔΔGs) for alanine substitutions and other residue substitutions was performed using Rosetta version 2.3 [26], Rosetta version 3.12 [46], and FoldX version 4 [57]. Benchmarking of computational alanine scanning predictive performance was performed using a subset of the AB-Bind dataset [58] that contains alanine point substitutions with quantified experimental ΔΔG measurements and known wild-type complex structures (347 mutants and ΔΔG values). A larger set with all point substitutions (including non-alanine substitutions) was also tested (531 mutants and ΔΔG values). Pearson correlation coefficients (PCC) between measured and predicted ΔΔG values, and receiver operating characteristic area under the curve (AUC) values for prediction of hotspot residues (measured ΔΔG for alanine residue substitution > 1 kcal/mol), were calculated using scipy and scikit-learn (sklearn) Python libraries, respectively.

Rosetta 2.3 ΔΔG calculations were performed using the "interface" protocol [26,59]. An example command line is:

```
rosetta.mactel -interface -intout pdb.ddgs.out -ignore_unrecogni-
zed_res -safety_check -skip_missing_residues -mutlist pdb.muts.txt
-extrachi_cutoff 1 -ex1 -ex2 -ex3 -constant_seed -jran 12 -yap -s
input.pdb
```

The input files specified on the command line denote the input PDB file ("input.pdb") and the list of mutations ("pdb.muts.txt"). The default protocol only models the mutant residue for ΔΔG calculation ("Ros2.3_norepack" in S2 Table), and additional flags were used on the command line to perform minimization of mutation-proximal side chains ("-min_interface -int_chi" flags; "Ros2.3_minint_chi" in S2 Table), minimization of mutation-proximal side chains and backbone ("-min_interface -int_bb -int_chi" flags; "Ros2.3_minint_bb_chi" in S2 Table), and rotamer-based packing of mutation-proximal side chains ("-repack" flag, "Ros2.3_repack" in S2 Table).

Rosetta 3 ΔΔG calculations were performed with two available computational mutagenesis protocols. One Rosetta 3 computational alanine scanning protocol was downloaded from a public resource containing benchmarks and Rosetta tools [60], and represents a separate implementation of the Rosetta 2.3 mutagenesis protocol noted above [26,59]. This protocol was recently used to predict TCR-peptide-MHC interface ΔΔG values [61]. In addition to the default protocol that does not repack neighboring side chains ("Ros3_norepack" in S2 Table), we also tested this protocol with repacking of neighboring side chains ("Ros3_repack" in S2 Table).

An example command line for running this protocol is:

```
rosetta_scripts.linuxgccrelease -s input.pdb -parser:protocol
alascan.xml -parser:view -inout:dbms:mode sqlite3 -inout:dbms:-
database_name rosetta_output.db3 -no_optH true -parser:script_vars
pathtoresfile = input.resfile chainstomove = 1,2 -ignore_zero_occu-
pancy false
```

We additionally performed alanine scanning using the Flex ddG protocol, which was developed recently in Rosetta 3 [62]. This protocol uses the backrub algorithm [63] to sample protein backbone conformations at the interface. We tested two sets of ΔΔG scores that are output by Flex ddG, representing different scoring functions reported by the authors [62]; they are shown as "flex_ddG-fa_talaris2014" and "flex_ddG-fa_talaris2014-gam" in S2 Table.

An example command line used for Flex ddG calculations in this study is:

```
rosetta_scripts.linuxgccrelease -s input.pdb -parser:protocol
flexddg.xml -parser:script_vars chainstomove = 1,2
mutate_resfile_relpath = input.resfile number_backrub_trials = 35000
max_minimization_iter = 5000 abs_score_convergence_thresh = 1.0 back-
rub_trajectory_stride = 7000 -restore_talaris_behavior -in:file:
```

```
fullatom -ignore_unrecognized_res -ignore_zero_occupancy false -ex1
-ex2
```

Prior to running ΔΔG calculations in Rosetta for alanine and non-alanine substitutions, antibody-RBD complex structures were pre-processed using Rosetta's FastRelax protocol [47], using the FastRelax flags noted above, to perform constrained backbone and side chain minimization to resolve unfavorable energies and anomalies that would bias energetic calculations, and to normalize such effects due to the differing resolutions of the experimentally determined structures.

For point substitution ΔΔG calculations in FoldX [57], complex structures were pre-processed using the FoldX RepairPDB command, and ΔΔG values were calculated using the FoldX PSSM command. Calculations of ΔΔGs for multiple substitutions were performed using the FoldX BuildModel command (using PDB files that were pre-processed by RepairPDB), followed by the AnalyseComplex command; reported ΔΔG values represent the mean ΔΔG from five simulations. FoldX version 4 was used for all FoldX simulations.

In the small number of cases where a variant RBD residue was not present in an experimentally determined structure, that structure was not included in the ΔΔG calculations for that residue and in the corresponding figure (**Figs 7**, **S6** or **8**). Those antibodies (and residues) are: C110, C135, S2H13, Sb23 (residue 477); C110, C135, S2H13 (residue 478); C135 (residue 484).

## Sequence conservation

Assessment of sequence conservation of SARS-CoV-2 RBD positions in the SARS-CoV-1 sequence was performed using SARS-CoV-2 (GenBank: QHD43416) and SARS-CoV-1 (GenBank: AAP13441) spike reference sequences aligned with BLAST [64]. The epitope residues of each antibody were defined as any SARS-CoV-2 residue within 5 Å of any antibody residue. An in-house Perl script was used to analyze SARS-CoV-2 antibody-antigen interfaces and calculate epitope conservation.

## Figures

Figures of structures were generated using PyMOL version 1.8 (Schrodinger, Inc.). Boxplots and dendrograms were generated using the ggplot2 [65] and factoextra [66] packages in R, and heatmaps were generated using the ComplexHeatmap package [67] in R.

## Supporting information

**S1 Table. Antibody-spike and antibody-RBD complex structures analyzed in this study.**
(DOCX)

**S2 Table. Performance of computational alanine scanning ΔΔG prediction for antibody-antigen interfaces.**
(DOCX)

**S3 Table. Comparison of ΔΔG predictions with measured monoclonal antibody neutralization of SARS-CoV-2 variants from Wang et al. [29].**
(DOCX)

**S4 Table. Comparison of ΔΔG predictions with measured monoclonal antibody neutralization of SARS-CoV-2 variants from Planas et al. [31].**
(DOCX)

**S1 Fig. Comparison of structural determination methods.** (A) Resolution, (B) interface buried surface area (BSA), and (C) number of interface atomic contacts between antibody and

RBD within a 5 Å distance cutoff were compared for structures obtained by cryo-EM and X-ray diffraction. Structures containing antibodies and nanobodies were separated to avoid possible bias in interface size due to smaller size of nanobodies. Statistical significance (Wilcoxon rank-sum test) between properties of cryo-EM and X-ray antibody-RBD structures is indicated at top (*: $p < 0.05$; **: $p < 0.01$; ***: $p < 0.001$). Due to small number of values for nanobody cryo-EM complex structures (N = 3), statistical comparisons were not performed for the nanobody-containing structures.
(PDF)

**S2 Fig. Examples of co-clustered antibodies, based on antibody RMSD, with shared RBD binding modes.** Shown are (A) antibodies MR17 (PDB code 7C8W) and 298 (PDB code 7K9Z), which have a 4.7 Å heavy chain orientation RMSD, (B) antibodies SR4 (PDB code 7C8V) and Sb16 (PDB code 7KGK), which have a 1.2 Å heavy chain orientation RMSD, and (C) antibodies BD-368-2 (PDB code 7CHF) and P2B-2F6 (PDB code 7BWJ), which have a 5.2 Å heavy chain orientation RMSD. The antibody-RBD structures are superposed by RBD (gray), and antibody chains (heavy, light, or nanobody) are colored separately as indicated.
(PDF)

**S3 Fig. Heatmap of antibody-RBD contacts, with the full set of 139 contacted RBD positions.** Labels and annotations are in accordance with the corresponding labels/annotations in Fig 2, and antibodies (rows) and RBD positions (columns) are ordered by hierarchical clustering in R. "BBClass" denotes the antibody classification from a previous study [23], with "ND" (empty cell) indicating that the class for the antibody was not described in that work. Antibodies in the heatmap are separated by the four major hierarchical clusters, which are labeled on left.
(PDF)

**S4 Fig. Antibody hierarchical clustering bootstrap confidence values.** Multiscale bootstrap resampling was performed in pvclust [49] in R, with the antibody-RBD contact data and 10,000 replicates. Values at each node denote the Approximately Unbiased (AU) bootstrap confidence, and red boxes delineate the four major clusters noted in this study, labeled accordingly.
(PDF)

**S5 Fig. Principal component analysis of antibody-RBD residue footprint data.** The x and y axes represent the first two principal components (PC1, PC2), with percentage of data variance represented by each principal component shown in parentheses. The 70 antibodies are shown as points, with colors and shapes representing Clusters 1–4, which were determined by hierarchical clustering analysis of antibody-RBD contact profiles. Selected points representing antibodies that are located on the periphery of cluster distributions are labeled by corresponding antibody names.
(PDF)

**S6 Fig. Computational assessment of antibody and ACE2 receptor ΔΔG values for RBD variants using FoldX.** FoldX [57] was used to simulate and compute binding affinity changes (ΔΔGs, in units of kcal/mol) for RBD point substitutions in 70 antibody-RBD complex structures and the ACE2-RBD complex structure (PDB code 6LZG). ΔΔG values for each RBD substitution are shown as a separate boxplot, with antibodies grouped by contact based cluster (1–4). The ACE2 ΔΔG value for each RBD point substitution is shown as a horizontal bar.
(PDF)

## Acknowledgments

Computing resources from the University of Maryland Institute for Bioscience and Biotechnology Research High Performance Computing Cluster were used in this study.

## Author Contributions

**Conceptualization:** Rui Yin, Brian G. Pierce.

**Data curation:** Rui Yin, Johnathan D. Guest, Ghazaleh Taherzadeh, Ragul Gowthaman, Ipsa Mittra, Jane Quackenbush, Brian G. Pierce.

**Funding acquisition:** Brian G. Pierce.

**Investigation:** Rui Yin, Johnathan D. Guest, Ghazaleh Taherzadeh, Ragul Gowthaman, Brian G. Pierce.

**Supervision:** Brian G. Pierce.

**Writing – original draft:** Rui Yin, Brian G. Pierce.

**Writing – review & editing:** Rui Yin, Johnathan D. Guest, Ghazaleh Taherzadeh, Ragul Gowthaman, Brian G. Pierce.

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
