## [Decision Letter · Decision Letter 0]

27 May 2021

Dear Dr. Pierce,

Thank you very much for submitting your manuscript "Structural and energetic profiling of SARS-CoV-2 antibody recognition and the impact of circulating variants" for consideration at PLOS Computational Biology.

As with all papers reviewed by the journal, your manuscript was reviewed by members of the editorial board and by several independent reviewers. In light of the reviews (below this email), we would like to invite the resubmission of a significantly-revised version that takes into account the reviewers' comments.

We cannot make any decision about publication until we have seen the revised manuscript and your response to the reviewers' comments. Your revised manuscript is also likely to be sent to reviewers for further evaluation.

Sincerely,

Charlotte M Deane

Associate Editor

PLOS Computational Biology

Thomas Leitner

Deputy Editor

PLOS Computational Biology

Reviewer's Responses to Questions

**Comments to the Authors:**

Reviewer #1: In this manuscript, Yin et al. perform epitope interaction analysis across a set of 70 solved 3D antibody:SARS-CoV-2 RBD structures, with the aim of predicting the likely impact of circulating variants on antibody recognition. First antibody “footprints” were derived through antigen contact analysis, and four broad epitope clusters were found through unsupervised clustering of these footprints. Mutational alanine scanning was then performed on each complex to identify key attractive binding interactions, and the impact of the specific mutations present in some SARS-CoV-2 viral variants were also assessed to deduce whether they are likely to have arisen to evade immune protection.

Though the authors’ approach for discretising SARS-CoV-2 epitope regions via. clustering antibody footprints is novel, by this stage in the pandemic the conclusions of this part of the work are largely confirmatory (see Barnes et al. [https://doi.org/10.1038/s41586-020-2852-1], Dejnirattisai et al. [https://doi.org/10.1016/j.cell.2021.02.032], and Niu et al. [https://doi.org/10.3389/fimmu.2021.647934]). The more impactful aspect of this study is that the authors proceeded to study the conservation of binding interactions within these epitope clusters, establishing a computational framework by which the energetic impact of viral variants can be predicted for the antibodies within. Effective computational workflows for this purpose will be important to rapidly highlight novel variants as “of concern” and to inform future vaccination strategies.

The paper is well-constructed, and an excellent level of detail is supplied in the methods to ensure reproducibility. I have a few minor concerns/suggestions for alterations to the paper:

1. The manuscript should be clearer that both X-ray crystal structures and cryo-EM structures are being analysed side-by-side in this dataset of 70 structures (perhaps in the main text and as an additional column to Table S1).

In my experience, cryo-EM structures tend to offer good information about approximate antibody binding region but can understate the number of interactions holding the complex together as side-chain information is often not well resolved. Were the cryo-EM structures on average of lower resolution? Was “score” required more often to fill-in side chains for the cryo-EM structures than for the XRD structures? I would be interested to see some summative statistics across the set of cryo-EM and XRD structures (perhaps split by antibody/nanobody) to capture whether fewer contacts/interactions or lower binding site interaction energies tend to be found in cryo-EM structures. If this is the case, please could the authors comment on whether the resulting sparsity of recorded interactions has led/could lead to uncertainty or errors in binding cluster assignment, particularly at the periphery of region definitions?

2. I think figure 5 would be augmented with a profile of ΔΔG of each RBD position for the binding of ACE-2 to the RBD (just as the authors did for figure 2). This goes also for figure 7/S4/S5, where the predicted ΔΔG impact of each mutation on ACE-2 binding would provide a relevant reference point to determine whether the mutations might have dual purpose for improving ACE-2 binding while escaping neutralising antibodies, improve ACE-2 binding while not escaping neutralising antibodies, or might even be deleterious to ACE-2 binding but even so be accommodated due to the benefit of immune evasion.

3. Relatedly, a timely addition to the paper would be analysis of the new "Indian" variant of concern (B1.617). What is the predicted impact of E484Q on cluster 2 antibodies, compared with E484K? And on ACE-2 binding? L452R, which was previously in the SI, could be brought into the main manuscript in accordance with its rising profile as a feature of the B1.617 variant. The authors could also consider analysing the combined affect on ΔΔG of multiple mutations (i.e. E484Q+L452R) in addition to analysing the mutations separately.

4. As the study is limited to analysis of antibodies against the receptor binding domain, I think the title should read “...SARS-CoV-2 receptor-binding domain antibody recognition...”. Many studies are now showing antibodies can engage SARS-CoV-2 in many other regions.

Typos

Line 118: I think this should be a to reference Figure S2 instead of S1.

Line 120: "Visualization the"

Line 135: "most are predicted block ACE2"

Line 175: "lusters"

Line 311: "which identifies and antibody-RBD structures"

Reviewer #2: It is vitally important that we understand the molecular interactions that underpin SARS-CoV-2 antibody-mediated neutralisation, and predict the pathways by which SARS-CoV-2 may escape immunity. In this study, Yin et. al. draw upon the wealth of Spike RBD-antibody structures to provide an integrated perspective on SARS-CoV-2 neutralisation. This work has two major values. First, through functional/structural classification the authors identify 4 clusters of anti-RBD antibodies; this provides a framework to understand and evaluate the ever-increasing list of anti-Spike mAbs/nanobodies. Secondly, by examining the theoretical resilience of antibody binding to mutation, the authors investigate the impact of on going, and future, antigenic variation. They also identify a particular cluster of antibodies that may withstand mutational escape and, therefore, would be a desirable specificity in vaccine-induced immunity. This work is of immediate value and importance, providing clarity on Spike-antibody interactions. I support the publication of this manuscript, however, there are some revisions that, in my opinion, would significantly improve the study.

Major Comments

• The categorisation and analysis of RBD-mAb interactions allow the authors to make predictions on the antigenic escape of certain mAbs, particularly in the context of recently emergent SARS-CoV-2 variants. It would be desirable to experimentally test a few of those predictions, this could be achieved with pseudovirus and synthesised antibodies. However, it may also be sufficient to verify predictions using analogous datasets in publications from others.

• A great value of this work is the analytical summary and organisation of a lot of important information regarding spike mAbs. Consequently the reader can gain a broad perspective of spike-antibody interactions. However, I think the authors are underselling their work by not communicating clearly. This is particularly the case in the figures where there is inadequate description or depiction of their findings. I provide some suggestions below.

• The discussion has a little too much focus on what could be done in the future to expand on this work. This somewhat detracts from the intrinsic value of the existent work presented in the manuscript. The discussion should be rebalanced to included more interpretation and contextualisation of the study alongside the literature.

Figure revision suggestions:

Figure 1: Maybe a few more representative superpositions of antibodies (in the supplementary material?). Also the dendogram is hard to read (e.g. the dashed line is not clear), can the branch lengths be expressed on a log scale to reveal the finer details of clustering? Also, where example structures are given highlight them on the dendogram to aid comprehension.

Figure 2: Is there value in including the BBClass? I appreciate it is relevant, but could also be covered in a supplement? It has little biological relevance.

Figure 3: This could be a lot clearer. A lot of information is expressed and it is hard to pick it apart. I would recommend multiple panels delivering discrete messages: 1) separate ACE2-RBD structure (i.e. without annotation) to orientate the reader (maybe a miniature inset of entire spike complex for context) 2) 4 example RBD-fAb structures illustrating the 4 clusters of antibodies (maybe context image of full spike in supplement to illustrate up/down conformation etc.) 3) structures annotating multiple antibodies (similar to what is already provided). Also, consistent color-coding of antibody clusters across figures is desirable. Clear communication in this figure will really aid the reader’s appreciation of the work.

Figure 4/5: The order of residues along the bottom is different to that in figure 2; is there a reason for this, can they be made consistent, again to ensure clarity?

Figure 6: this would be complemented with an RBD structure color-coded by conservation, alongside a structure annotated by antibody cluster for context.

In my opinion, review and revision of the figures with a view on clarity of message will increase the impact of this work.

Minor Comments:

• Line 96 states 8 angstrom cutoff whereas figure 1 legend states 7.

• Line 119-120 needs rewording.

• Related to line 122 – I think more could be done to draw a relationship between the RMSD analysis (F1) and clustering (F2), might color coding help here?

• Line 127 – Figure S2 should be S3.

• Line 175 – “lusters”.

• Line 204, erroneous parentheses.

• Line 310-311, erroneous “and”.

Reviewer #3: The authors performed structural and energetic analysis of SARS-CoV-2 S-RBD/antibody recognition after classifying antibodies by RBD binding residues using unsupervised clustering. Based on the structural data, they also evaluated the capacity of prevalent viral variant mutations to disrupt antibody recognition. Their study suggested that the cluster-2 antibodies were capable of targeting escape antibodies. The analysis should be able to serve as a useful reference for vaccine and therapeutic design. The computational work is solid and comprehensive though lacking experimental data to support some conclusions, e.g., cluster-2 antibodies are able to effectively neutralize escape SARS-CoV-2 variants while antibodies in other clusters are likely to be affected by these mutations. I recommend to publish this work after a few typos or minor errors are corrected.

1. Line 194-195, B.1.525 (E484K), and a recently reported variant of concern, B.1.526 (E484K). why B.1.525 and B.1.526 are the same? Do the authors mean L452R in B.1.526?

2. Line 311, delete the ‘and’ between identifies and antibody-RBD.

**Have the authors made all data and (if applicable) computational code underlying the findings in their manuscript fully available?**

Reviewer #1: Yes

Reviewer #2: Yes

Reviewer #3: Yes

PLOS authors have the option to publish the peer review history of their article (what does this mean?). If published, this will include your full peer review and any attached files.

Reviewer #1: No

Reviewer #2: No

Reviewer #3: No
---

## [Decision Letter · Decision Letter 1]

25 Aug 2021

Dear Dr. Pierce,

We are pleased to inform you that your manuscript 'Structural and energetic profiling of SARS-CoV-2 receptor binding domain antibody recognition and the impact of circulating variants' has been provisionally accepted for publication in PLOS Computational Biology.

Best regards,

Charlotte M Deane

Associate Editor

PLOS Computational Biology

Thomas Leitner

Deputy Editor

PLOS Computational Biology

Reviewer's Responses to Questions

**Comments to the Authors:**

Reviewer #1: I thank the authors for fully addressing each of my suggestions in their revised version of the manuscript.

Reviewer #2: I thank the reviewers for responding to my suggestions in a positive manner. The manuscript is improved and I support publication in its current form. A valuable contribution, thank you.

Reviewer #3: none

**Have the authors made all data and (if applicable) computational code underlying the findings in their manuscript fully available?**

Reviewer #1: Yes

Reviewer #2: Yes

Reviewer #3: Yes

PLOS authors have the option to publish the peer review history of their article (what does this mean?). If published, this will include your full peer review and any attached files.

Reviewer #1: No

Reviewer #2: No

Reviewer #3: No

---

## [Editor Report · Acceptance letter]

2 Sep 2021

PCOMPBIOL-D-21-00728R1 

Structural and energetic profiling of SARS-CoV-2 receptor binding domain antibody recognition and the impact of circulating variants

Dear Dr Pierce,

I am pleased to inform you that your manuscript has been formally accepted for publication in PLOS Computational Biology. Your manuscript is now with our production department and you will be notified of the publication date in due course.

With kind regards,

Andrea Szabo
